# Non-disruptive collagen characterization in clinical histopathology using cross-modality image synthesis

Adib Keikhosravi[1,2,7], Bin Li [1,2,3,7], Yuming Liu[2], Matthew W. Conklin [4], Agnes G. Loeffler[5] & Kevin W. Eliceiri [1,2,3,6✉]

The importance of fibrillar collagen topology and organization in disease progression and prognostication in different types of cancer has been characterized extensively in many research studies. These explorations have either used specialized imaging approaches, such as specific stains (e.g., picrosirius red), or advanced and costly imaging modalities (e.g., second harmonic generation imaging (SHG)) that are not currently in the clinical workflow. To facilitate the analysis of stromal biomarkers in clinical workflows, it would be ideal to have technical approaches that can characterize fibrillar collagen on standard H&E stained slides produced during routine diagnostic work. Here, we present a machine learning-based stromal collagen image synthesis algorithm that can be incorporated into existing H&E-based histopathology workflow. Specifically, this solution applies a convolutional neural network (CNN) directly onto clinically standard H&E bright field images to extract information about collagen fiber arrangement and alignment, without requiring additional specialized imaging stains, systems or equipment.

[1] Department of Biomedical Engineering, University of Wisconsin-Madison, Madison, WI, USA. [2] Laboratory for Optical and Computational Instrumentation, University of Wisconsin-Madison, Madison, WI, USA. [3] Morgridge Institute for Research, Madison, WI, USA. [4] Department of Cell and Regenerative Biology, University of Wisconsin-Madison, Madison, WI, USA. [5] Department of Pathology, MetroHealth Medical Center, Cleveland, OH, USA. [6] Department of Medical Physics, University of Wisconsin-Madison, Madison, WI, USA. [7] These authors contributed equally: Adib Keikhosravi and Bin Li. ✉email: eliceiri@wisc.edu

Collagen forms the structural network of the extracellular matrix (ECM) in biological tissues and is the most abundant protein in vertebrates. The organization of fibrillar collagen, such as fiber density, distribution, and alignment are important tissue characteristics and have been demonstrated to be critical factors involved in a wide array of diseases. Properties of collagen fiber organization have been identified as candidate image biomarkers in a number of pathological studies including cancer, aging, wound healing, atherosclerosis, and diabetes. In cancer, Tumor Associated Collagen Signatures (TACS) were described as alterations in collagen reorientation and deposition during mouse mammary tumor progression[1]. Subsequently, it was observed that TACS-3, where highly aligned collagen fibers are oriented perpendicular to the tumor boundary, was negatively prognostic in human breast cancer[2]. Similar correlations were found in other cancer types such as skin[3], ovarian[4], prostate[5], pancreas[6,7], and others[8].

Collagen fibers in tissues can be visualized by several methods: (1) Stains: Movat's pentachrome and Masson's trichrome and antibody staining[9]; (2) Polarization based microscopy, in which picrosirius red (PSR) is usually used to enhance detection due to low retardance of native collagen[10], or polarized imaging of native collagen using LC-PolScope and Polychromatic PolScope[11,12]; (3) Second Harmonic Generation imaging (SHG)[13–15] which is now the gold standard in many stromal research studies.

SHG imaging is a label-free imaging technique highly specific to collagen fibers. Fibrillar collagen has a non-centrosymmetric structure which is necessary for producing detectable SHG signals. Because of the specificity, label-free detection, and the ability to penetrate deep into thick tissues, SHG imaging has become a widely used tool for investigating collagen topology and organization and it is considered a candidate quantitative imaging method for visualization of collagen fibers in histopathology studies. However, it is not used clinically due to the high cost, complexity, and the typical requirement for optics experts to operate. Undoubtedly, SHG imaging has great research advantages including large imaging depth, optical sectioning, and the ability to provide higher-order information such as forward SHG to backward SHG ratio and polarization-based excitation[4,15,16], However, most of the tissue sections used in clinical histopathological studies for collagen topology and organization investigation have the standard thickness of 5 μm, which makes the mentioned advantages of SHG imaging not of primary interest,[2,6,7,17–19]

We showed that LC-PolScope, a sensitive polarization imaging system, can be used for imaging collagen in histopathology slides without the need for intensifying birefringence using picrosirius red staining, with results comparable to SHG when quantifying fiber orientation or alignment[11,20], Although LC-Polscope is simpler and very cost effective compared to SHG imaging, and does not require additional staining methods, this modality requires several modifications to the pathologist's microscope including additions of a chromatic filter, special variable retarders, and circular and linear polarizers. The final image is rendered computationally after post processing. This adds more steps thus making it less suitable for incorporation into real-time pathology workflows.

Over the last decade, cross-modality image synthesis has attracted many researchers in the field of image analysis. The intent of their research has mainly been to synthesize radiative image modalities such as Computed Tomography (CT) scans from non-radiative modalities such as Magnetic Resonance Imaging (MRI) images[21,22], or even more radiative and more expensive modalities such as Positron Emission Tomography (PET) from CT-scans[23]. These image synthesis methods can be roughly categorized in three groups: (1) Segmentation based methods, which segment the tissue into different classes and assign new contrast values based on known attenuation properties[24,25]; this type of method may fail due to the existence of some ambiguous tissue classes, such as air and bone. (2) Atlas based methods that register the subject specific scan to an atlas and warp the attenuation map of the atlas to the subject, in which image synthesis accuracy highly depends on the registration accuracy[26,27], (3) Machine learning based methods, in which a trainable model such a Convolutional Neural Network (CNN), Gaussian Mixture Model (GMM), or other method is used to learn the relationship between MRI and CT images,[28,29]

Deep learning has been extensively used in biomedical image analysis applications from single image super-resolution techniques[30], image segmentation[31,32], image classification[33–35] and more. Deep Convolutional Neural Networks (CNN), inspired by human visual cortex neuronal structures, have outperformed most of the previous computer vision and classification methods, which are based on human defined features such as SURF features[36]. The expression "digital pathology" was coined when referring to advanced slide-scanning techniques in combination with AI-based approaches for the detection, segmentation, scoring, and diagnosis of digitized whole-slide images[37]. Xu et al. proposed a novel GAN-based approach to convert the H&E staining of WSIs to virtual immunohistochemistry staining based on cytokeratins 18 and 19, an approach that potentially obviates the need for destructive immunohistochemistry-based tissue testing[38]. Ilhe et al. used Cycle-GAN for segmenting unlabeled data of VGG cells, VGG Cells dataset, bright-field images of cell cultures, a live-dead assay of C. Elegans and X-ray-computed tomography of metallic nanowire meshes[39].

In this study, we developed a CNN model to generate synthesized collagen images from bright field (BF) images of H&E stained histopathology slides such as those regularly used in diagnostic pathology. This model was trained on an image dataset consisting of more than one million pairs of collagen and BF images of human breast and pancreatic cancer. Known prognostic biomarkers such as fibrillar collagen orientation and alignment were measured and compared on an independent validation set for both real and synthesized SHG images of fibrillar collagen. No significant difference was found between results obtained from the real and synthesized collagen images. Our proposed CNN model achieved superior performance compared to two other state-of-the-art Generative Adversarial Networks (GAN) for image to image translation and cross-modality image synthesis. Such a low cost, instantaneous and highly accurate deep learning-based computational method can open the gate for discovery and application of new biomarkers, such as stromal collagen signatures, in pathologic diagnosis, thereby improving risk stratification and the formulation of precision treatment plans for individual patients.

## Results

**Visualization of collagen image and stromal biomarkers.** After training the network, we first evaluated the synthesis performance on some of the images from two testing sets. Figure 1 compares the real and synthesized collagen image of a pancreatic cancer TMA core. The top row shows a BF image (Fig. 1a), synthesized collagen (Fig. 1b), and the real collagen image (Fig. 1c) of the same core. The bottom row shows the same image types from an area of the same core (blue rectangle) at higher magnification (Fig. 1d–f). The synthesized collagen shows smoother fiber ridges compared to the real collagen image of SHG imaging. This alteration can actually facilitate fiber tracking in CurveAlign[40]. This discontinuity is due to unmet phase matching conditions in SHG imaging that we will address in the discussion.

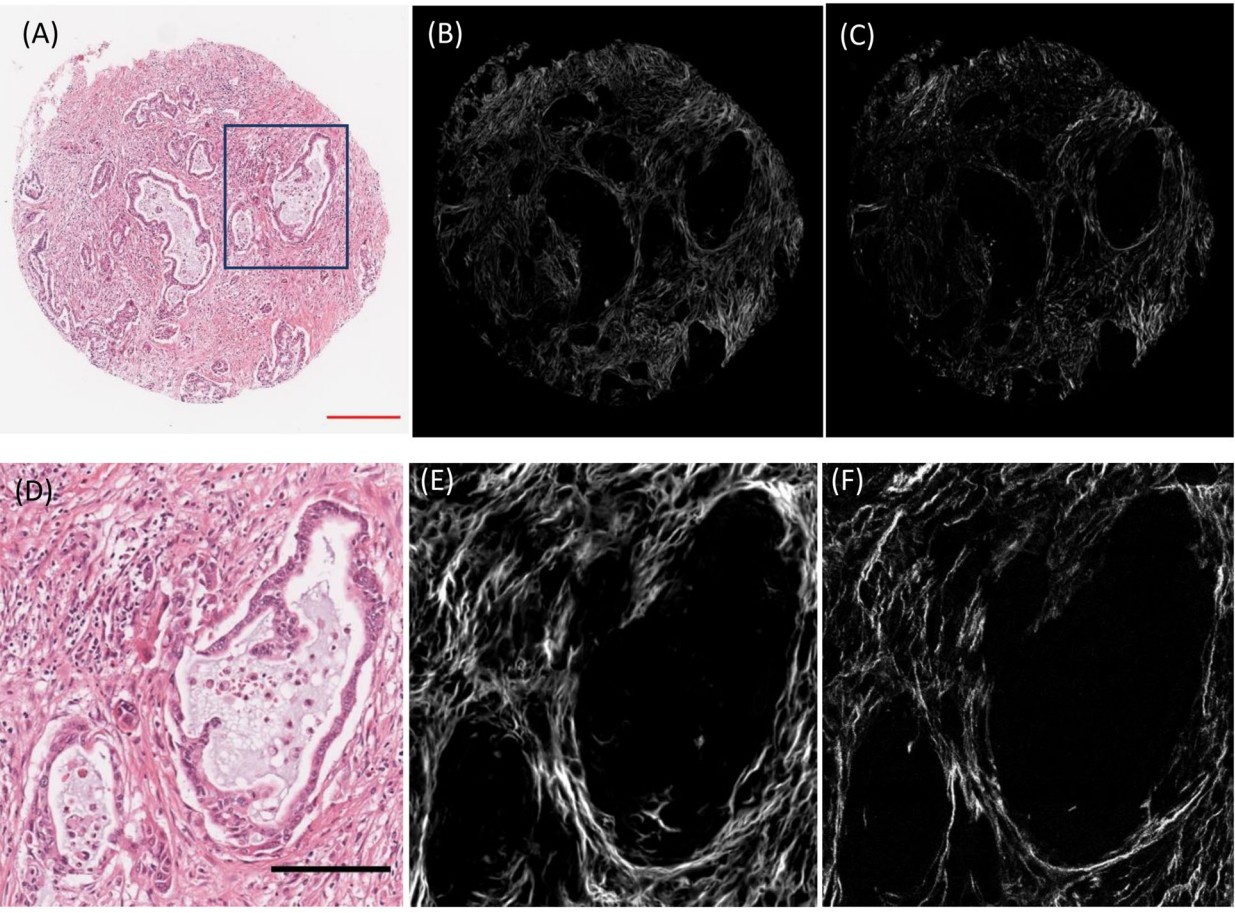

**Fig. 1 Comparison of real and synthesized collagen images of a pancreatic cancer TMA core.** Top row shows a BF image of pancreatic ductal carcinoma (**a**), synthesized collagen using our proposed CNN network from the BF image (**b**), and the ground truth collagen image (**c**). Bottom row shows the zoomed in version of the same images as top row from the blue ROI rectangle (**d**–**f**). Scale bars: Red 200 μm and Black 100 μm.

Tumor associated collagen signatures are prognostically linked to poor survival among patients with breast cancer[1,2], TACS-3 is defined as highly aligned collagen fibers close to the tumor interface that have a large angle with respect to the tumor boundary. Figure 2 shows that the synthesized SHG image can be used to identify TACS-3 regions and yields similar relative angle measurements for a breast cancer TMA core previously annotated as TACS-3 by Conklin et al.[2] The first row shows the BF image of a whole TMA core (Fig. 2a), a heatmap of fiber orientations with respect to tumor boundaries on a synthesized collagen image (Fig. 2b), and the corresponding heatmap from the real collagen image (Fig. 2c). The second row shows the same characteristics in the region of interest depicted in the rectangle on the H&E stained image in the top row (Fig. 2d–f). The third row shows the fiber orientation colormap with respect to tumor boundaries (Fig. 2g) the boxplot for fiber angles for both synthesized (red) and real (blue) collagen images (Fig. 2h). In the fiber orientation heatmap, red shows angles of 60 degrees and above with respect to tumor boundaries, yellow shows 45–60,10–45 is shown in green, and angles of 0–10 are blank/clear.

**Output comparison to state-of-the-art networks**. To further validate the synthesized SHG images generated by our trained model, we calculated image similarity metrics such as Peak Signal-to-Noise Ratio (PSNR), Structural Similarity Index Measure (SSIM)[41], Mean Square Error (MSE) and $l_{1,norm}$ between network output and ground truth images using an independent new

pancreatic cancer TMA that was not used in the training dataset (Table 1).

Figure 3 shows sample collagen image patches synthesized from BF images by our network in comparison to two state of the art networks: Pix2Pix[42], proposed for image to image translation, and Cycle-GAN[43], proposed for cross-modality image synthesis. Pix2Pix produced similar but blurry results compared to real collagen images, but Cycle-GAN failed to retrieve the collagen context from BF images. Our network showed superior performance for generating a synthesized collagen image. In fact, it even removed the shot noise and overcame the phase-matching constraint that results in a pixelated image in actual SHG imaging,[13,16]

**Collagen fiber quantification results**. Collagen reorganization parameters such as fiber orientation and alignment have been extensively explored and widely accepted as hallmarks of disease progression and patient prognosis through alteration in tumor cell signaling with the surrounding microenvironment[2,6,14,18,44], Thus, it was important to verify that real and synthesized collagen images can statistically achieve the same result with regard to fiber orientation and alignment. We used a new independent pancreatic TMA[7] (not in the training set) for verification. Collagen and BF images were registered using the same algorithm described in the methods section and synthesized collagen images were obtained from BF images, so both synthesized and real collagen images have the same size. Real collagen images were divided into a grid of 256 × 256 blocks and those with less than

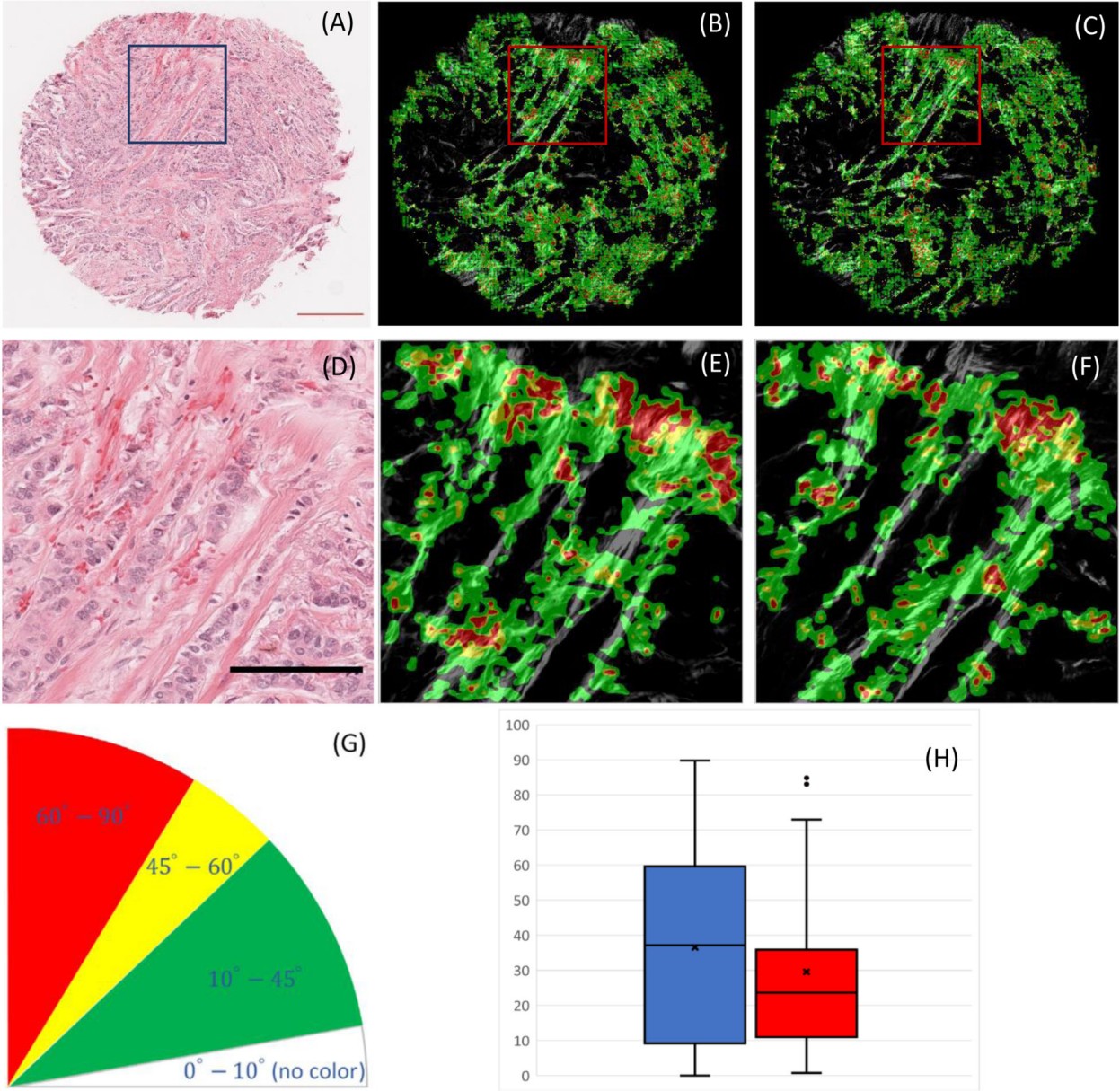

**Fig. 2 Synthesized SHG image and actual SHG image can locate similar TACS-3 regions and yield similar relative angle measurements for a breast invasive ductal adenocarcinoma TMA core previously used by Conklin et al.** The top row shows a BF image (**a**), synthesized collagen image using our proposed CNN network from the BF image overlaid by fiber orientation analysis results from CurveAlign (**b**) and ground truth collagen image overlaid by fiber orientation analysis results from CurveAlign (**c**). Middle row shows the zoomed in version of the same images as the top row from the region of interest in the blue rectangle (**d**–**f**). Bottom row shows the fiber orientation colormap with respect to tumor boundaries (**g**) the boxplot for fiber angles for both synthesized (blue) and real (red) collagen images (**h**) (*P* < 0.005). Scale bars: Red 200 µm and Black 100 µm.

30% second harmonic signal were excluded from analysis (An example of selected ROIs is illustrated in Fig. 4a, b). This resulted in 5840 image blocks for both real and synthesized collagen images. Images were first analyzed using CT-FIRE fiber segmentation software[40] and then CurveAlign[45] was used to calculate fiber orientation and fiber alignment for each block. The comparison.

A Bland-Altman plot shows the difference between two measurements of the same parameter vs. the average of these values, which can be used to describe the agreement between two quantitative measurements by constructing limits of agreement[46,47], Bland-Altman plots of sinusoids of orientation and alignment for both image sets are plotted in Figs. 4c, d. Blue dotted lines show the mean of differences or constant bias

**Table 1 Performance comparison among proposed CNN, Pix2Pix and Cycle-GAN.**

|                   | PSNR      | SSIM     | MSE       | $l_{1.norm}$ |
|-------------------|-----------|----------|-----------|--------------|
| Proposed network  | **32.11** | **0.66** | **0.002** | **0.022**    |
| Pix2Pix           | 27.26     | 0.59     | 0.003     | 0.036        |
| Cycle-GAN         | 23.72     | 0.46     | 0.007     | 0.055        |

Metrics: Peak Signal-to-Noise Ratio (PSNR), Structural Similarity Index Measure (SSIM)[41], Mean Square Error (MSE) and $l_{1,norm}$. Bold: Best values.

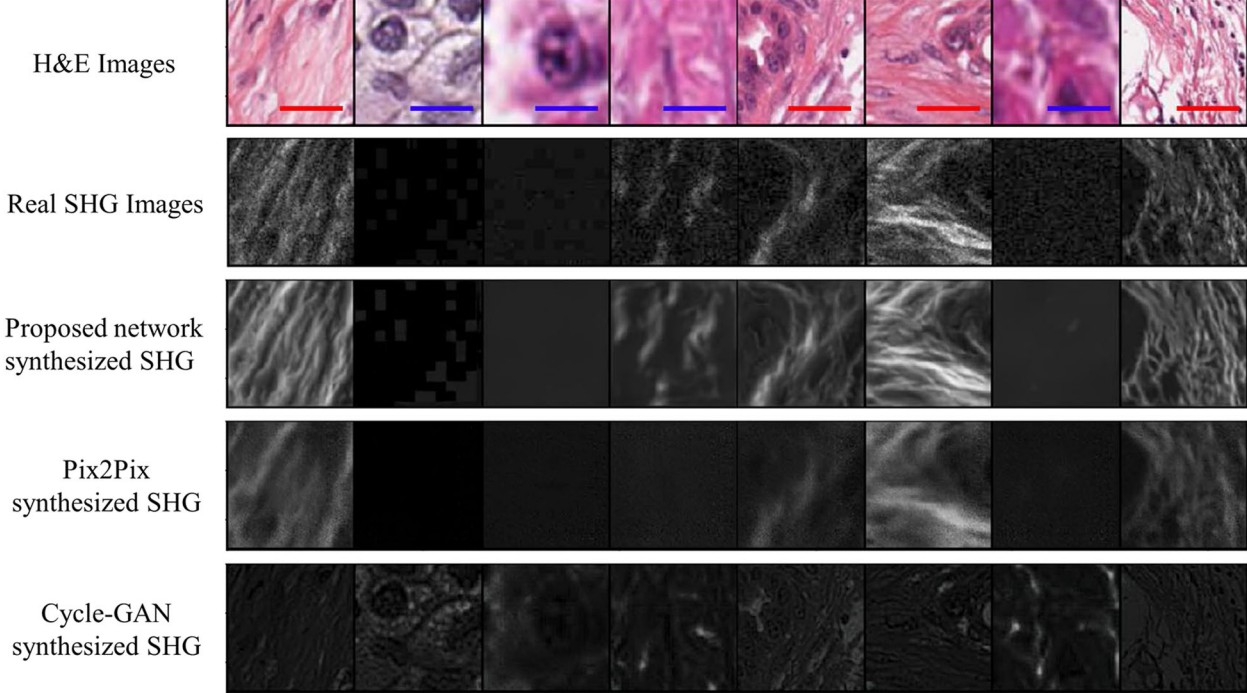

**Fig. 3 Comparison of synthesized collagen images generated by our proposed network and two other image synthesis networks used for cross-modality image synthesis and image to image translation.** Scale bars: Red 20 μm and Blue 10 μm.

between two measurements and the red lines show the 95% confidence interval (CI) (mean ± 1.96 STD). The bias between our measurements are negligible (0.02 for orientation and −0.07 for alignment), which suggests no systematic difference between measurements from real and synthesized collagen images of SHG microscopy. For both orientation and alignment, differences between two measurements of more than 95% of the data falls between ±1.96 STD, which is proposed as a good agreement between two methods by Bland and Altman[46].

The correlation coefficient ranges from −1 to +1, where values of −1 or +1 indicate a perfect negative or positive linear relationship, while value of 0 indicates no linear relationship. According to the guidelines for interpretation[48], the agreement between synthesized and real collagen images for orientation (0.86) and alignment (0.78) is excellent

## Discussion

A variety of methods have been used for imaging fibrillar collagen for biological and pathological research, including different stains, immunohistochemical dyes, polarized and SHG imaging etc., but none of these methods have been incorporated in the workflows of diagnostic pathology. The role of stroma as a key player in disease progression is only slowly being acknowledged in the clinical domain. Stromal biomarkers are still not well recognized, and pathologists are not yet trained to use them in prospective outcome studies, let alone standard clinical practice. In addition, the methods traditionally used to study and characterize collagen have not been amenable for use in routine practice, as they are costly and require equipment and technical expertise not available in most diagnostic laboratories.

Many studies have by now demonstrated that the amount of collagen deposition, fiber orientation around the tumor, and alignment of collagen fibers are prognostic biomarkers[2,6,7,18], Accurate measurement of these parameters will be an important factor in prospective clinical studies and clinical applications. Picrosirius red stained tissue imaged with polarized microscopy is

one of the widely used imaging methods in stromal analysis research. The source of contrast in this imaging method is enhanced dichroism detected through birefringence measurement using crossed polarizers. The intensity of transmitted light passed through two polarizers at various angles can be calculated using Malus' law:

$$I = I_0 \, cos^2\theta \qquad (4)$$

where $I$ is the intensity of light passed through the second polarizer (analyzer), $I_0$ is the intensity of an incident beam of linearly polarized light, and $\theta$ is the azimuthal angle between incident light polarization and transmission axis of the analyzer[49]. This shows the limitation of using picrosirius red combined with crossed polarizers, which results in intensity change due to change in collagen orientation and total extinction when the optical axis of the collagen fiber is perpendicular to the analyzer axis. This in turn results in discontinuities and can result in false fiber segmentation. We have previously shown that using fluorescence imaging of picrosirius red stained tissue can help solve this problem, however, this type of imaging is not used in routine diagnostic pathology, and an additional staining protocol other than H&E is required[50].

Similar to polarized microscopy, generating second harmonic signal, which is specific to non-centrosymmetric molecules such as collagen I, II, and III is highly orientation dependent. This means that SHG signal is maximum for zero degrees and minimum for 90 degrees of angle between light polarization and fiber angle[13,16], Moreover, due to the small size of the individual collagen fibrils (less than 50 nm), usually there are so many fibers in one focal volume, and recorded signal is emanated from all of the fibrils inside the focal volume (the SHG signal from one fibril is too weak to detect)[51]. Two fibrils with opposite polarities generate π-phase-shifted signal, that completely cancel each other. As SHG results from the coherent summation of several fibrillar responses in every pixel, the relative polarity between adjacent fibrils can highly affect the signal intensity. Therefore, the tissue is defined by the following parameter f that indicates the number of fibrils

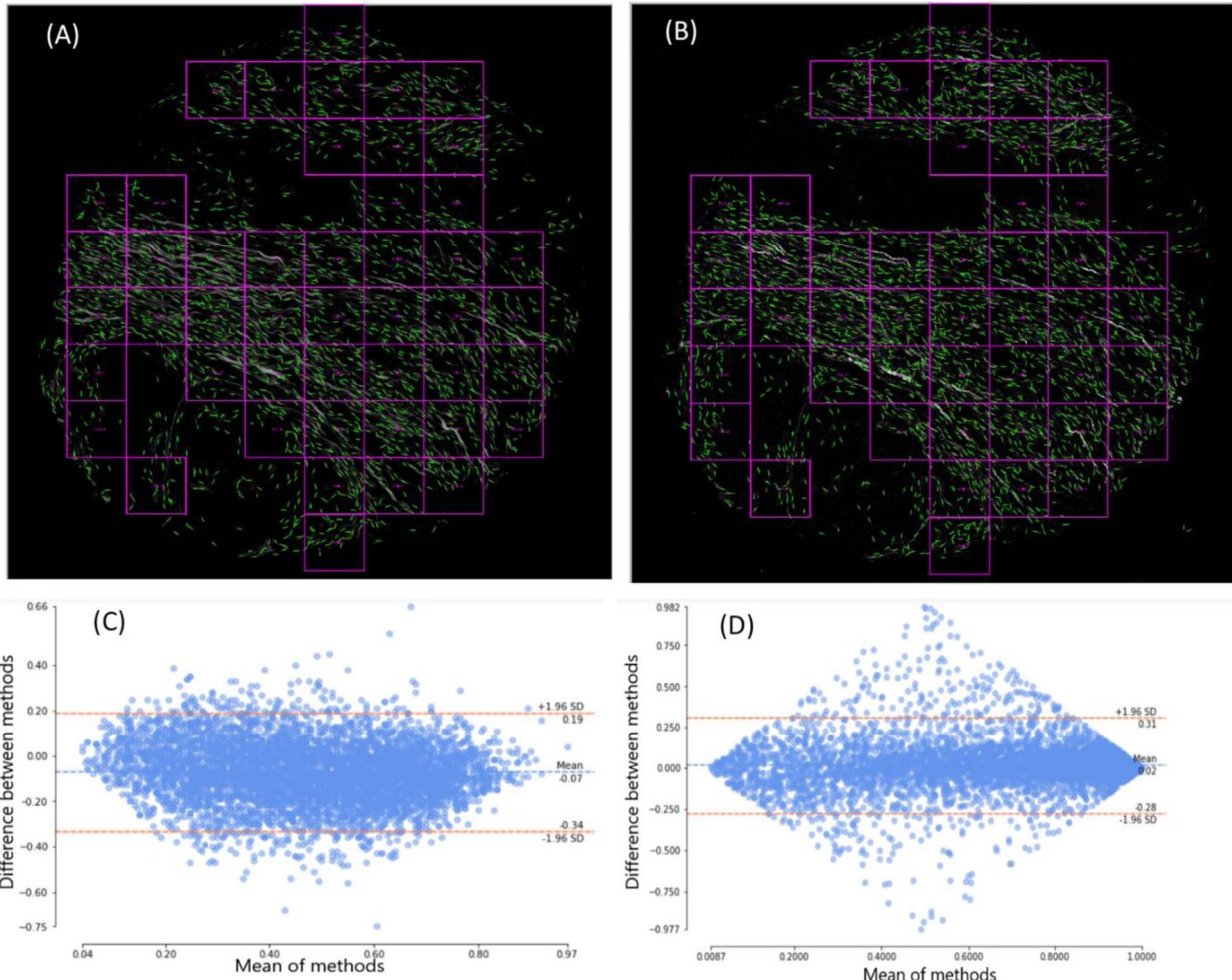

**Fig. 4 No significant difference in fiber orientation and alignment were found between synthesized SHG images and real SHG images of an independent pancreatic TMA slide.** Top row shows the ROI selection for fiber metrics analysis based on 30% SHG signal for synthesized SHG (**a**) and real SHG (**b**). Bottom row shows the Bland-Altman analysis plot for fiber alignment inside ROI as the difference of two measurement vs. the mean (**c**) and the same plot for sin() of orientation of the fibers inside blocks (**d**). Each block is 256 × 256 pixels (110 × 110 μm).

with positive polarity:

$$f = \frac{N(+\chi^2)}{N(+\chi^2) + N(-\chi^2)} \qquad (5)$$

Where $\chi^2$ is the second order susceptibility and $N(\pm\chi^2)$ is the total number of collagen fibril dipoles with one polarity or the other[51]. The SHG signal is zero for $f = 0.5$, because of equal number fibrils with opposite polarities, and it increases as $f$ moves towards 0 or 1, which means one polarity is dominant. As a result, the SHG signal depends on both collagen presence and fibril polarity that can change the intensity and cancel the SHG signal in case of antiparallel alignment. This discontinuity can be seen in the SHG images of collagen, which again can result in low accuracy in pixel-wise fiber segmentation (Fig. 1).

Unlike polarization microscopy and SHG imaging that exploit collagen optical properties for contrast, convolutional neural networks have to take a fundamentally different approach. CNNs consist of a trainable set of filters that are applied to an image to adaptively extract desired information by minimizing an objective loss of function based on a target output[52]. In the input BF image, Eosin stains the cytoplasm and other structures including extracellular matrix components such as collagen[53–55] in up to five shades of pink and red blood cells as intensely red[56]. The eosinophilic (substances that are stained by eosin)[53] structures in human tissue are generally composed of intracellular or extracellular

proteins. Therefore, the training task for the network is not merely color-based or even texture-based classification, but to learn where to expect collagen in tissue. Our CNN has learned this with a high level of accuracy. This result is very promising for two reasons: 1) by including more samples from different tissue types and retraining the network, CNNs can replace certain imaging methods, such as SHG and polarization, for thinly sectioned slides in research and the clinic; and 2) this network can be re-trained for other cellular and extracellular components stained with their specific dyes.

To train our CNN model, we used $l_1$ norm and SSIM with a factor $\rho$ that were changed during the course of the training. Using SSIM for training not only decreases the blurriness of the synthesized SHG but also keeps the main structures in the image and removes the shot noise that adversely affects SHG imaging. As can be seen in the figures provided, the synthesized collagen images have a smoother fiber representation compared to real collagen images from SHG microscopy, which are affected by imaging noise. Although GAN has been previously put forth as a solution for blurriness in reconstructed images, our method achieved superior performance for image-to-image translation in comparison to two state of the art GAN structures, Pix2Pix[42] and Cycle-GAN[43] (Table 1 and Fig. 3).

The blocks selected from our validation TMA were from collagen-rich regions of breast tissue and pancreatic tissue stroma, with more than 30% SHG signal that show different levels of

alignment visually. The agreement between real and synthesized collagen images for orientation (0.86) and alignment (0.78) shows that (1) BF images of H&E stained slides contain high levels of information in collagenous regions; and (2) powerful computational techniques such as deep learning-based algorithms can harness this information. Although the results show that our CNN can be used to analyze the organization of the collagen network, several sources contribute to deviation from complete correlation with actual collagen images. These include: (1) SHG shot noise and lack of detectable SHG signal can result in fragmented fibers from a single fiber, whereas synthesized SHG shows a smoother fiber ridge that can be easily detected as an intact single fiber. (2) The appearance of collagen fibers in the actual SHG image highly depends on the working numerical aperture (NA) of the optical system, meaning that images acquired with 20x, 0.75 NA are not the same as down-sampled versions of images acquired with 40x, 1.25 NA. Moreover, due to photon density needed for SHG imaging of fibrillar collagen, very small fibers won't be detected; or if these small fibers are close enough, they will be represented as a single fiber. Whereas this is not the case in BF imaging as it is a first order optical phenomenon.

In this study, we showed the ability of our designed convolutional neural network to synthesize a nonlinear microscopic modality, specifically fibrillar collagen images (produced by SHG imaging), from BF images of H&E stained slides. These images can be used to extract quantitative stromal biomarkers that are not detectable by routine pathologic examination but appear to have prognostic significance. There are slight variations in H&E staining in different pathology laboratories, and the colors could be additionally distorted by the WSI modality used to create a digitized image. However, these differences are not significant enough to hinder the network in its ability to identify collagen and its arrangement relative to the tumor. We are modifying and training the same designed network to normalize color and fix staining issues in H&E stained samples that will be the subject of future work. Moreover, although in this study we have only used breast and pancreatic tissue, the network should be readily adaptable to other disease types, as well. We used greater than one million image patches to achieve the high accuracy and robustness reported here, but we believe the performance of this network can be even more improved by including more training images from other tissue types. We are actively engaged in this exercise. This trained model is available on GitHub[57] will soon be available in FIJI[58], open source software platform for biological image analysis and can be tested with different disease types.

## Methods

**Histological samples.** Breast tissue slides were acquired through various collaborations. All slides were de-identified. The cohorts are as follows: core needle biopsies (CNB) from healthy volunteer women; CNBs from prophylactic mastectomies of women at high risk for the development of breast cancer; diagnostic breast biopsy tissue with benign or normal results; and CNBs from the breast contralateral to cancer which was without malignancy or atypia, with paired tissue from the tumor itself and an adjacent normal area from the diseased breast. The biopsies were done while the breast was still in place but just before the surgery started—the patient was anesthetized. In total we used pairs of collagen (SHG) and BF images of annotated regions of 278 normal and 211 breast cancer tissue. The pancreatic tissue dataset consisted of 1196 biopsy cores of tissue microarray (TMA) slides that had been used in our previous studies. Complete information about the samples can be found in[6,7,10]; one additional TMA slide named PA 2081b, was purchased from US Biomax, Inc. website (www.biomax.us) and SHG and BF imaging was performed as described in next section. All tissues were formalin-fixed and paraffin-embedded, then cut into 5 μm thin slices, affixed to a slide and stained with hematoxylin and eosin (H&E) before mounting with a coverslip.

**Imaging systems.** Bright-field imaging of pancreatic samples was performed on all the TMA slides using an Aperio CS2 Digital Pathology Scanner (Leica Biosystems) at 40x magnification, and all the cores were separated manually. All the SHG imaging and bright-field imaging of breast samples in this study was done with a

custom built integrated SHG/bright field imaging system. A MIRA 900 Ti: Sapphire laser (Coherent, Santa Clara, CA) tuned to 780 nm, with a pulse length of less than 200 fs, was directed through a Pockels cell (ConOptics, Danbury, CT, USA), half and quarter waveplates (ThorLabs, Newton, NJ, USA), beam expander (ThorLabs), a 3 mm galvanometer driven mirror pair (Cambridge, Bedford, MA), a scan/tube lens pair (ThorLabs), through a dichroic beam splitter (Semrock, Rochester, NY) and focused by either a 40X/1.25NA water-immersion or 20X/0.75NA air objective lens (Nikon, Melville, NY). SHG light was collected in the forward direction with a 1.25 NA Abbe condenser (Olympus) and filtered with an interference filter centered at 390 nm with a full width at half maximum bandwidth of 18 nm (ThorLabs MF390–18). The back aperture of the condenser lens was imaged onto the 5 mm aperture of a H7422-40P photomultiplier tube (Hamamatsu, Hamamatsu, Japan) the signal from which was amplified with a C7319 integrating amplifier (Hamamatsu) and sampled with an analog to digital converter (Innovative Integration, Simi Valley, CA). Timing between the galvo scanners, signal acquisition, and motorized stage positioning was achieved using our custom acquisition software called WiscScan. Bright field images of breast samples were captured with the same system using a MCWHL2 white LED lamp (ThorLabs) set up for Koehler illumination. White light from this lamp was separated from SHG light traveling through the condenser assembly using a short pass dichroic mirror with a cutoff at 670 nm (Semrock). An RGB CCD camera (QImaging QICAM, Surrey, BC, Canada) was used to capture bright field images through WiscScan to allow for acquisition within a single application. Both SHG and white light images were tiled with 5% overlap using automation provided by WiscScan. Stage positions for individual images and pixel size data were stored in Bio-Formats image metadata[59] and this was then used by the grid/collection stitching ImageJ plugin to reassemble a high-resolution large field of view image of the entire imaged area.

For the pancreatic TMAs, BF and SHG imaging was performed on every single core. For breast samples, suitable imaging areas were identified by eye using the bright-field imaging mode, and the captured image array size was typically ~1.5 mm × 1.5 mm. In order to alleviate out of focal plane issues due to the unevenness of the tissue slice, 3 z-planes were captured per SHG image and then maximum-intensity projected to capture the entire axial field of view. The SHG and bright-field images were registered using in-house written code[60]. Both images were then used to perform a segmentation of the epithelium from the rest of the tissue. This segmentation was turned into a binary mask for tumor boundary creation.

**Image preprocessing.** After acquisitions, BF and SHG images of collagen were stitched using a stitching plugin in FIJI[58]. Since SHG and BF images are not registered during the acquisition, we developed an image registration algorithm to overlay BF and SHG images of H&E stained slides. A two-step process was used to register SHG and BF images, the details of which have been previously described[60]. However, we briefly overview the general scheme of the algorithm here. The first step was to extract the stroma, which is stained as red using eosin from H&E stained slides using K-means clustering for generating a pseudo-collagen image. The second step was using an iterative intensity-based image registration algorithm to find the affine transform for mapping the pseudo-collagen image to the SHG image. Here we briefly explain each step:

*Stroma extraction from H&E stained tissue image.* H&E staining protocol paints the nuclei dark blue using hematoxylin, while eosin stains the cytoplasm and other structures, including extracellular matrix components such as collagen[53–55], in up to five shades of pink and red blood cells as intensely red[56]. We first adjusted the dynamic range of each channel of the BF image by adjusting the mean and standard deviation of the histogram of each channel; then, using HSV color space, we separated the stroma from the nuclei. Nuclear images were smoothed using a Gaussian filter, and binarized. Since cytoplasm also stains with eosin but is not present in SHG images of fibrillar collagen, we dilated this binary image to create a cell mask. By element-wise multiplication of this mask with the stroma image we could remove the cytoplasm from the stroma image. The resulting image mainly contains collagen ($I_{BF,C}$).

*Registration based on mutual information maximization.* The algorithm is multi-resolution and starts with the coarsest level of the images and uses the joint probability distribution of a sampling of pixels from two images to measure the certainty that the values of one set of pixels map to similar values in the other image. In another word, the algorithm will try to find the optimal affine transform parameters that maximize the mutual information between template (SHG) and source image (BF) defined as:

$$M\left(I_{SHG},\ T\left(I_{BF,C}\right)\right) = \sum_{x \in X} \sum_{y \in Y} p(x,y) \log\left(\frac{p(x,y)}{p(x)p(y)}\right) \tag{1}$$

Where $p(x,y)$ is the joint probability mass function of SHG image ($I_{SHG}$) and affined transformed of extracted collagen image form FB image ($T\left(I_{BF,C}\right)$). $p(x)$ and $p(y)$ are the marginal probability mass functions of $I_{SHG}$ and $T\left(I_{BF,C}\right)$, respectively. $x$ are the pixel values of SHG image ($I_{SHG}$) and $y$ is the affine transformed of the pseudo-collagen image extracted from BF image $T\left(I_{BF,C}\right)$. The

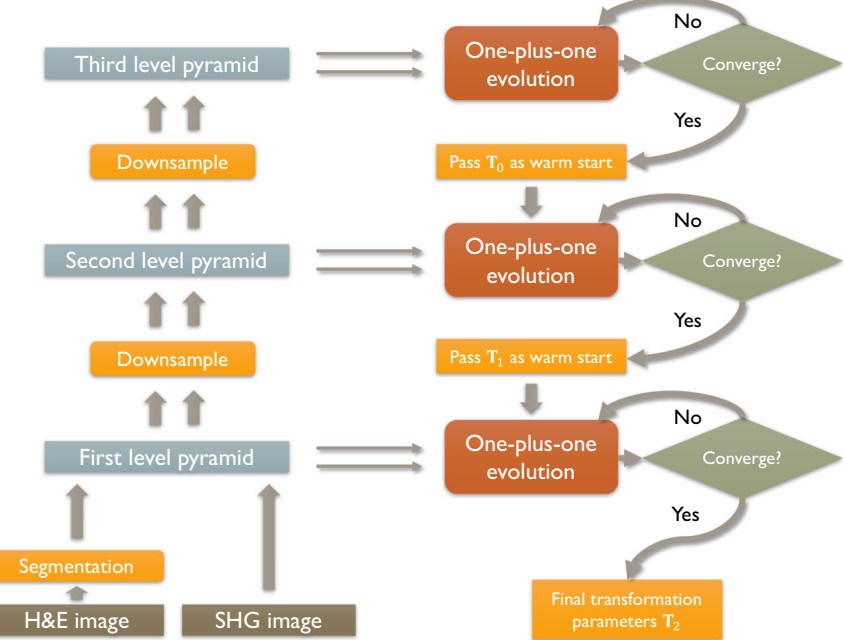

**Fig. 5 Block diagram of the multi-resolution registration algorithm.** BF image is first segmented to extract the ECM image features using K-means color segmentation scheme. The ECM image and SHG image are then downsampled to the coarsest level of the image registration pyramid to find the best parameters of the fitted affine transform by maximizing mutual information using one-plus-one evolutionary algorithm.

marginal and joint probability densities of the image intensities are estimated using Parzen windowing. At each iteration, the pseudo-collagen image extracted from BF image is rotated, translated or scaled based on affine transform parameters and mutual information between this transformed image and collagen image is calculated. The transformation parameters found at lower resolution are passed to the higher resolution level as an initial estimate in an image pyramid, until the optimal solution is found for the finest level. This process will be continued using one-plus-one evolutionary algorithm until the convergence criterion is met that means the mutual information is maximized. Figure 5 shows the block diagram of this process.

After registering BF and SHG images, for every pair of these images, we used both original and down-sampled images by a factor of 2 before generating image patches. This method not only increases the size of the dataset, but also forces the network to learn the underlying feature space at lower resolutions that makes the network's output robust to different imaging setups. Image patches were initially generated with size of $100 \times 100$ pixels with 50% overlap between images and then rescaled to $128 \times 128$ before training. More than one million pairs of input-output patches were created using this method.

**Network design**. We initially started the project by testing various state-of-the-art networks that have been proposed for image synthesis and segmentation. However, most of these networks either failed to converge at all or produced blurry results. Two of the networks that could successfully converge the images, Cycle-GAN[43] and Pix2Pix[42], are represented in the results section for comparison. Based on our previous study in which we tested the performance of state-of-the-art networks for grading pancreatic cancer using both cell and stromal features, ResNet 101[61] outperformed all other networks in classifying all image patches, even those containing only stroma. For this study, an encoder-decoder network was designed consisting of residual modules in both the encoder and decoder. The encoder estimates the underlying feature space by passing the input into a stack of residual modules. The decoder structure is similar to that of the encoder, however despite residual modules in the encoder that reduce the spatial size of the feature map by using a stride of 2, the encoder up-samples these feature maps to finally half the size of the output. A pixel shuffle layer as the last layer will reconstruct the network output to the size of input-output pair ($128 \times 128$). The proposed CNN structure is shown in Fig. 6 and its constructing blocks are described below:

*Residual blocks.* Residual mapping was previously introduced for image representation in dictionary learning[62,63], and it's a powerful shallow representations for image retrieval and classification[64,65], This concept has been used later for training deep residual networks to overcome the vanishing backpropagated gradient problem in ResNet networks[61]. If H(x) is the desired underlying mapping, we let the stack of convolutional layers in each residual module estimate another mapping, F (x) = H(x) – x, so the original mapping will be H(x) = F(x) + x. It has been previously shown in dictionary learning and deep learning that optimizing residual

mappings are easier than optimizing original mappings[66]. In this study, we found that a combination of residual mapping with two different filter sizes and skip connections yielded better results for edge preserving for SHG image synthesis. In the encoder of the model, every residual block is composed of two convolutional layers with the same number of kernels each followed by a batch normalization and a leaky ReLU layer. The first convolutional layer has a filter size of $3 \times 3$ with a stride of 2 for reducing the size of feature maps, and the second convolutional layer has a filter size of $1 \times 1$ and a stride of one (Fig. 7). Output from this channel is added to the output of the input passed through a convolutional layer with filter size of $2 \times 2$ and stride of 2 with the same number of kernels.

*Skip connections and up-sampling.* Skip connections added to the network effectively improved the training accuracy and speed. These paths are composed of two convolutional layers followed by batch normalization and leaky ReLU layers, with filter sizes of $3 \times 3$ and $1 \times 1$ and strides of 2 and 1, respectively. The decoder will recover the underlying feature space with the same size as the encoder by bicubic interpolation at each step that can be connected using skip connections.

*Pixel shuffle layer.* The purpose of this layer is to recover the full-size synthesized collagen image ($128 \times 128$) from $64 \times 64$ feature space produced by the last layer of the decoder. To this end, we used a pixel shuffle layer similar to the one proposed in[64]. The upscaling of the last feature maps to the size of the collagen image was implemented as a convolution with a filter $\theta_{sub}$ whose stride is $1/r$ ($r$ is the resolution ratio between the feature maps and collagen image). Let the size of the filter $\theta_{sub}$ be $f_{sub}$. A convolution with a stride of $1/r$ on feature maps with a filter $\theta_{sub}$ (weight spacing $1/r$) would activate different parts of $\theta_{sub}$ for the convolution. The weights that fall between the pixels will not be activated. The patterns are activated at periodic intervals of $mod(x, r)$ and $mod(y, r)$ where $x$ and $y$ are the pixel position in the collagen image. This can be implemented as a filter $\theta_{final}$, whose size is $n \times r^2 \times f \times f$, given that $f = f_{sub}/r$ and $mod(f\_sub, r)=0$. This can be written as

$$Y = \gamma \left( \theta_{final} \times Y + b \right) \qquad (1)$$

Where $\gamma$ is periodic shuffling operator to rearrange $r^2$ channels of the decoder output to the size of collagen image. Figure 8 illustrates operating mechanism of pixel shuffle layer visually for the case where the upscaling factor is 2 and the kernel size is $4 \times 4$. The upsampling procedure can be considered as convolving the kernel with a subpixel image which is created by zero-padding unpooling with a stride of 2. The purple kernel weights are set first and activated by nonzero pixels. Then by moving one subpixel to the right in the subpixel image, the blue weights are activated, and so on for green and red weights (Fig. 8a). Instead of convolving the (1, 1, 4, 4) kernel with the unpooled subpixel image, the input can be convolved with the (4, 1, 2, 2) kernel directly, and using periodic shuffling we can achieve the same output as illustrated in Fig. 8b: Thus, pixel shuffle layer can upsample the feature maps with a much smaller memory footprint.

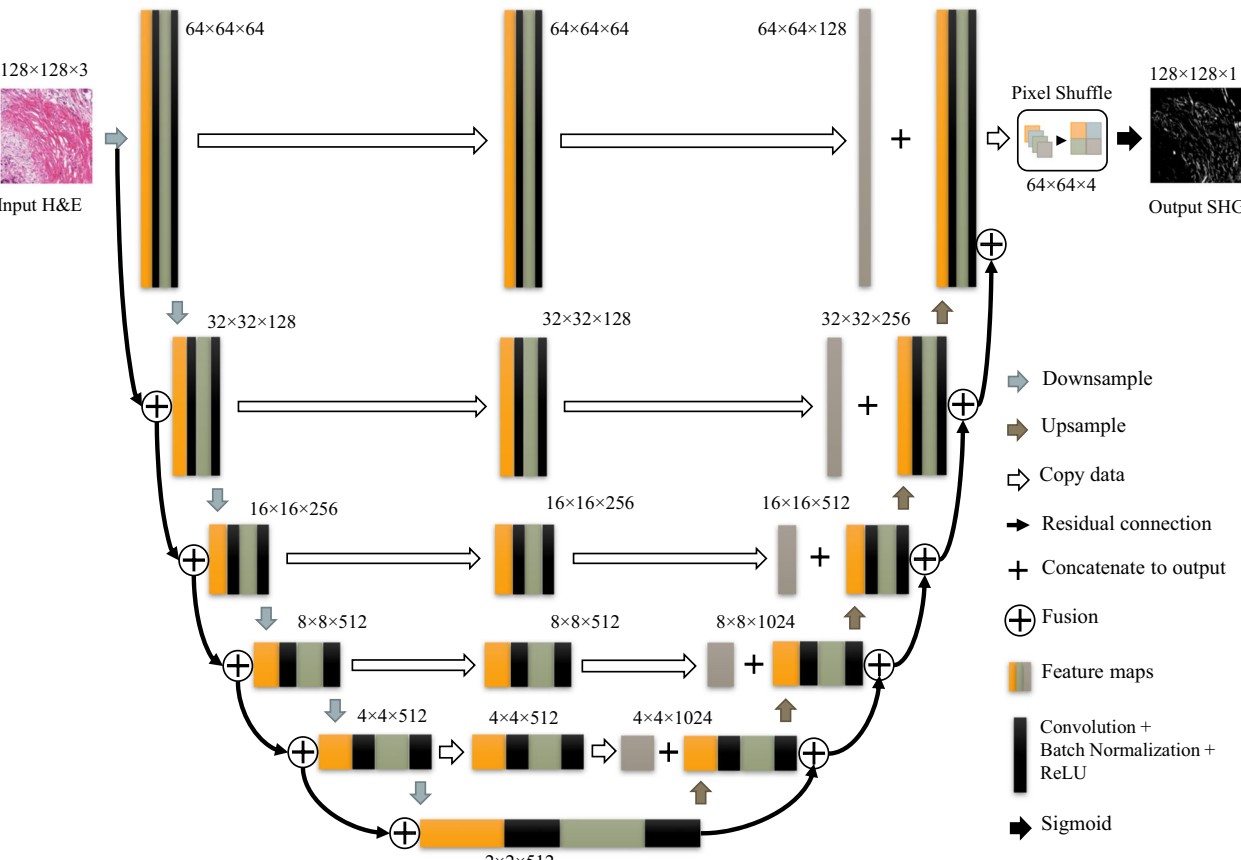

**Fig. 6 Proposed CNN structure for synthesizing SHG images of collagen from BF images of H&E stained slides.** An ecoder-decoder network with residual blocks and skipping concatenations. The last upsampling layer is replaced by a pixelshuffle layer. The number of channels and the size of the feature maps are listed next to each layer.

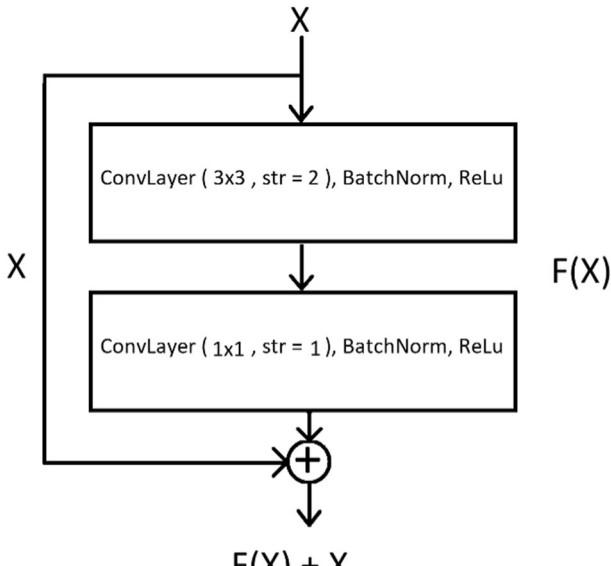

**Fig. 7 Residual module used in our proposed network.** The module consists of two convolutional layers followed by batch normalization and leaky ReLu layers, with filter sizes of $3 \times 3$ and $1 \times 1$ and strides of 2 and 1, respectively. Residual signals are expanded by a $1 \times 1$ convolution to match the number of channels of the output.

**Model training and loss functions**. The objective function for training the network is crucial in determining the quality of synthesized collagen images. Using only pixel-wise norms such as $l_1$ OR $l_2$ norms between the real and synthesized collagen image, which while easy to optimize, often causes blurriness and correlates poorly with human perception of image quality. This is because these norms return the average of several possible solutions, which does not perform well for high-dimensional data[67]. To address this issue, we trained our CNN using linear combination function of Structured Similarity Index Measure (SSIM) in addition to $l_1$ norm between the synthesized ($S$) and the real SHG image ($R$). SSIM can be calibrated to capture perceptual metrics of image quality. In addition, its pixel-wise gradient has a simple analytical form and is inexpensive to compute and therefore can be easily back-propagated in gradient descent algorithm. Let $x$ and $y$ be two patches of equal size from the two images $S$ and $R$ being compared. Assume $\mu_x \left( \mu_y \right)$ denote the mean, $\sigma_x^2 \left( \sigma_y^2 \right)$ denote the variance of the patch $x(y)$ respectively, and $\sigma_{xy}$ denote their covariance. Therefore, the SSIM function can be defined as:

$$SSIM(x, y) = I(x, y)^\alpha C(x, y)^\beta S(x, y)^\gamma \qquad (2)$$

where $I(x, y) = \frac{\left( 2\mu_x \mu_y + c_1 \right)}{\left( \mu_x^2 + \mu_y^2 + c_1 \right)}$ is the luminance based comparison, $C(x, y) = \frac{\left( 2\sigma_x \sigma_y + c_2 \right)}{\left( \sigma_x^2 + \sigma_y^2 + c_2 \right)}$ is a measure of contrast difference and $S(x, y) = \frac{\sigma_{xy} + c_3}{\sigma_x \sigma_y + c_3}$ is the measure of structural differences between the two images. $c_i$, for $i = \{1, 2, 3\}$, are small values added for numerical stability, and the $\alpha$, $\beta$ and $\gamma$ are the relative exponent weights in the combination. The structural similarity between the images $S$ and $R$ is averaged over all corresponding patches $x$ and $y$. This single-scale measure assumes a fixed image sampling density and viewing distance and may only be appropriate for certain range of image scales. Our final loss function is defined as:

$$L(S, R) = \rho l_{1, norm}(S, R) + (\rho - 1)SSIM(S, R) \qquad (3)$$

where $\rho$ is between 0 and 1. Since both terms in the objective are differentiable, we can train the neural network using gradient descent, adopting standard back propagation methods.

**Statistics and reproducibility**. Fiber orientation was defined as the angle with respect to the horizontal axis, which causes angle ambiguity between the angles

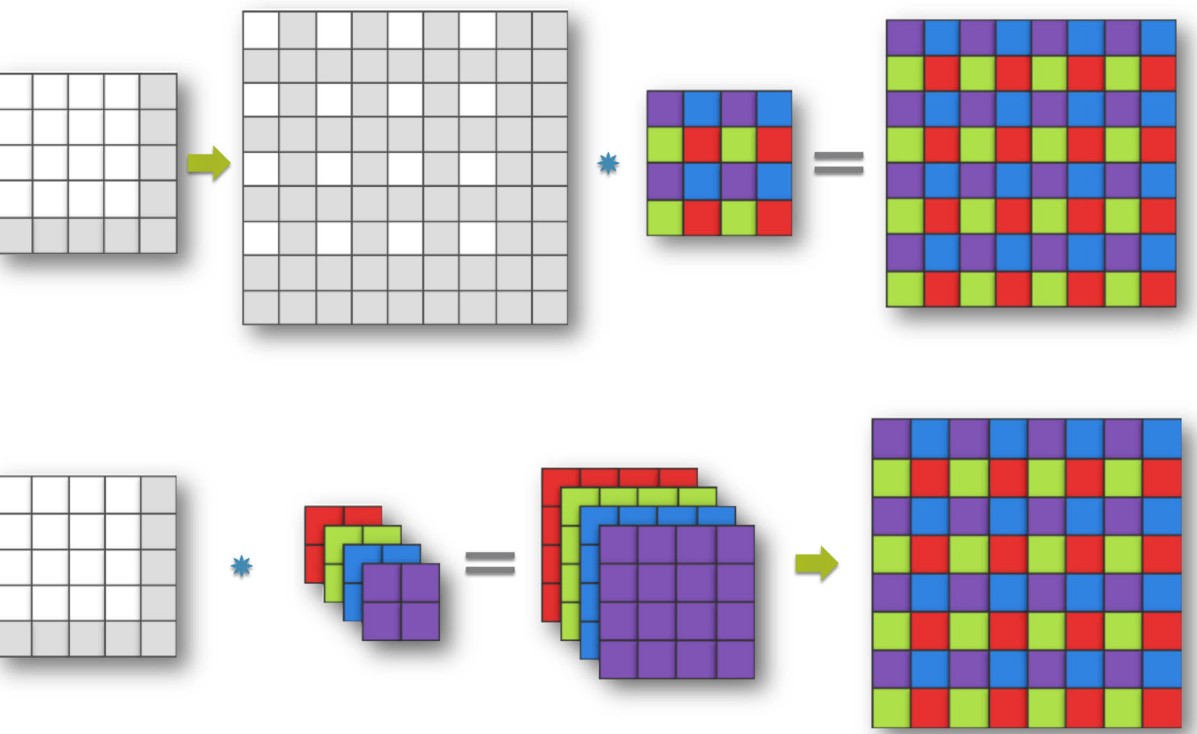

**Fig. 8 Sub-pixel convolution can be interpreted as group convolution + periodic shuffling.** Pixel shuffle layers upscale the feature maps directly in the input space and have a smaller memory footprint compared to transpose convolution and bilinear convolution upscaling.

around the lower and upper limits. More specifically, angles close to 0° and 180° essentially indicate similar orientations, but the absolute angle differences are significant. Hence a sinusoid function "$sin\ (\pi\ x/180)$" (x is the orientation value in degree) was used to map the orientation values from [0 180] degrees to [0, 1] to avoid the ambiguity. Collagen alignment coefficient ranging from 0.0 to 1.0 indicates how similarly the orientations of collagen fibers are distributed in a given area. It is defined as the mean resultant vector length in circular statistics[68], with 1.0 indicating all fibers are aligned in one direction, while small values close to 0.0 indicate fibers are oriented in random directions. Pearson correlation coefficient between synthesized and real collagen images was 0.86 for orientation and 0.78 for measured alignment. Pearson's correlation coefficient, which is a statistical measure of the strength of a linear relationship between paired data, is defined as:

$$\rho_{xy} = \frac{cov(x, y)}{\sigma_x \sigma_y}$$

Where $cov(x, y)$ is the covariance and $\sigma_x$ and $\sigma_y$ are the standard deviations of variables $x$ and $y$, respectively.

**Informed consent**. This study is human subject exempt as approved by the University of Wisconsin Institutional Review Board.

**Reporting summary**. Further information on research design is available in the Nature Research Reporting Summary linked to this article.

## Data availability
The datasets generated during and/or analyzed during the current study are available from the corresponding author on reasonable request. The dataset contains digital images of registered SHG and H&E image pairs in TIFF format. Part of the data is currently being analyzed for other studies and the access will be released in later GitHub repository updates. Data associated with Figs. 2 and 4 can be found in the Supplementary Data 1 and 2.

## Code availability
The code is available at GitHub code repository[57] and it is being maintained by the researchers at the Laboratory for Optical and Computational Instrumentation.

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

## Acknowledgements

We gratefully acknowledge funding from the Semiconductor Research Corporation (SRC), the Morgridge Institute for Research and the UW Carbone Cancer Center. This work was also supported by NIH grant R01CA199996.

## Author contributions

A.K., B.L. and K.W.E. conceptualized the technique and application. A.K., B.L. and K.W.E. designed the experiments. M.W.C. and A.G.L. performed the biology and pathology validation. A.K., B.L., Y.L. performed the data acquisition, developed the

code for post-processing and performed the data analysis. A.K., B.L. and K.W.E. wrote the paper with input from Y.L., M.W.C. and A.G.L.

## Competing interests

The authors declare no competing interests.
