## [Peer Review File · Communications Biology]

Reviewers' comments:

Reviewer #1 (Remarks to the Author):

The use of GAN (Generative Adversarial Networks) and recent deep learning techniques to map from one modality to another modality has been an active area of study in recent years. This paper utilizes a novel network architecture to map histology images from two types of cancer histopathology slides, breast cancer and pancreatic cancer, from standard H&E stains to SHG (Second Harmonic Generation). The use of machine learning for stromal collagen image synthesis allows for more extensive research on using collagen as markers for disease severity.

The paper addresses an important topic in the analysis of histological samples and the approach can be extended to other scenarios such as IHC (Immunohistochemistry). The paper is well motivated, and the writing is clear and cogent. I recommend that the paper to be accepted.

In the method section, the authors presented first a method to provide registration of the H&E stained images to SHG images so that matched training images can be generated. Over 1 million patches of H&E stain and SHG were used in the training. The residual map network architecture they developed was compared with Pix2Pix and CycleGAN. Their results were significantly better in PSNR, SSIM, and other metrics.

In summary, the paper is well written, with clear method descriptions which would allow for other researchers to reproduce the approach. The approach will be helpful to biology researchers in a wide variety of domains.

Suggested improvements:

Line 19: an extra period after the word workflow.

Line 220: One Plus One Evolution algorithm is not described clearly w/o citation. What were the parameters searched with evolution algorithm? The core algorithms of mapping are open sourced. How about the registration algorithm? If it is open sourced, then a citation to where it can be found would be very helpful.

Reference is well researched. There are some more recent work that would be helpful for the reader so the authors may consider adding them:

Bera, K., Schalper, K.A., Rimm, D.L. et al. Artificial intelligence in digital pathology — new tools for diagnosis and precision oncology. *Nat Rev Clin Oncol* 16, 703–715 (2019) doi:10.1038/s41571-019-0252-y

Xu, Z., Moro, C. F., Bozóky, B. & Zhang, Q. GAN-based virtual re-staining: a promising solution for whole slide image analysis. *ArXiv.org* <https://arxiv.org/abs/1901.04059> (2019).

Ihle, S.J., Reichmuth, A.M., Girardin, S. et al. Unsupervised data to content transformation with histogram-matching cycle-consistent generative adversarial networks. *Nat Mach Intell* 1, 461–470 (2019) doi:10.1038/s42256-019-0096-2

Reviewer #2 (Remarks to the Author):

This work demonstrates an automated image analysis methodology based on convolutional neural networks (CNNs) that provides information regarding collagen fiber organization and alignment in

stroma. This information is known to have prognostic value especially regarding different types of cancers (so-called tumor associated collagen signatures, TACS). Importantly, the demonstrated methodology can be applied on standard H&E stained tissue slides providing a simple optical methodology to transfer the utilization of this information towards clinical practises.

In overall, the manuscript proceeds logically and reads quite well. I'm also quite convinced of the presented results and think that this work presents a significant advance in the field. In order to further improve this high-quality work, I suggest the authors to consider the following points:

1. The sections regarding the algorithm and analysis details ('Registration based on mutual information maximization', 'Residual blocks' and 'Pixel shuffle layer') do not read as well as the other sections. I suggest to clarify and polish these sections further so that a reader would get a clearer picture of the presented approach, and would be able to better reproduce the results. It would also be useful to provide some references regarding the math related to Eq. (1).

2. The caption of Fig. 1. is not very clear/helpful to a reader. I suggest to clarify the overall workflow using a sentence or two.

3. Synthesis of second-harmonic generation (SHG) images is perhaps overly emphasized, because of which I fear that less careful readers may not fully appreciate the main novelty of this work. In my point of view, the important/interesting information that is extracted from the H&E images is related to the collagen organization. I understand that this information is often accessed by performing SHG microscopy, motivating to synthesize SHG images from the H&E images. However, from the biomedical point of view the SHG images as such are not important, rather the information they provide (in an easily accessible form). My suggestion is that the authors re-consider how much they want to emphasize the importance of SHG modality here. I think the impact of this work could be further increased by clarifying this point in the manuscript.

4. Line 169, Correct 'Kohler' to 'Koehler' or 'Köhler'.

5. Lines 187-188: I think a singular use of the reference [68] is adequate in this sentence.

6. Line 448: 'Collagen crystals', you mean highly-ordered collagen fibers etc.?

7. Line 473: Clarify to the readers what the values 0.86 and 0.78 are and signify in this sentence. I don't think their meaning is obvious.

8. Lines 490-495: Unnecessarily long sentence.

Reviewer #3 (Remarks to the Author):

The manuscript by Keikhosravi and coauthors presents a convolutional neural network (CNN) that can identify collagen fibers present in conventional H&E stained paraffin histology sections and then synthesize new images containing only the collagen fibers. Collagen images synthesized from the CNN are compared to true collagen maps produced via second harmonic generation (SHG) and shown to give an extremely high degree of agreement. Similarly, the model is compared to other approaches from the literature and shown to give improved results while suppressing noise inherent in the SHG process. Finally, the SHG and synthesized collagen maps are shown to give similar fiber orientation

maps, suggesting that the generated images from H&E slides could be used in diagnostics based on collagen orientation in place of more costly SHG.

From an engineering perspective the work and the presented images are impressive. The main limitation is that while CNN avoids the need for SHG imaging to evaluate collagen (which is extremely expensive) or specialized stains such as picosirius red, it still depends on having a high resolution whole slide image, or at least high resolution images of the entire diagnostically relevant regions of the tumor stroma. This is problematic, since at my center, images of breast and other surgical pathology specimens are never scanned, and in fact no whole slide imaging scanner is even present in the facility. If individual fields of tumor pathology could be selected, that would simplify data acquisition, but it would still force purchasing microscope hardware with the correct specifications (NA, sensor color filters, etc) to match the training set used on the CNN as well as training of personnel to select the correction locations to feed the CNN. This is at odds with the argument in the manuscript that the proposed CNN method could generate collagen alignment data without specialized equipment, at least as long as slide scanning remains uncommon. More likely, whole tissue slides would have to be sent to a third party service to be imaged on the correct hardware and then evaluated with the CNN. While slides are often sent to third parties to perform complicated or specialized diagnostics (e.g. FISH, etc) during routine workup of tumor specimens, the paper would benefit from a stronger argument that the CNN is the most practical approach compared to just having the third party do IHC or other special staining (or even SHG if only a small region of the tumor needed to be sampled).

Finally, one thing that was less clear to me is how the authors envision collagen imaging (or any kind) being integrated into clinical practice. Would this be something that a pathologist would be trained to interpret, similarly to how special stains are used today? Or would it be a purely digital method where a CNN generates the images, and then an additional algorithm grades the specimen? The latter would strengthen the argument for the CNN approach since it would anyway require whole slide imaging.

Minor:

Figure 4: Are the CurveAlign data here the orientations described in the methods section? If so, a colorbar with the scale would be helpful.

“CNBs from prophylactic mastectomies of non-diseased women”

What was being biopsied if the patient was receiving a prophylactic mastectomy? Or were the biopsies performed on a mastectomy specimen (after surgery)?

“A MIRA 900 Ti: Sapphire laser (Coherent, Santa Clara, CA) tuned to 780 nm, with a pulse length of approximately 100 fs”

The femtosecond model is the 900F, and I don't think it can produce a pulse as short as 100fs. The actual length is probably somewhat longer.

“an interference filter centered at 390 nm with a full width at half maximum bandwidth of 22.4 nm (Semrock)”

Should give the model number.

“imaged onto the 5 mm aperture of a 7422-40P photomultiplier tube”

The model is actually the “H7422-40P”.

“(An example of selected ROIs is illustrated in Figure 6A and 6B).”

Figure 6 panels should probably be labeled A, B, ... to remove ambiguity.

“Figure 5C and 5D.”

I think this actually meant to refer to Figure 6.

Reviewer #1 (Remarks to the Author):

The paper addresses an important topic in the analysis of histological samples and the approach can be extended to other scenarios such as IHC (Immunohistochemistry). The paper is well motivated, and the writing is clear and cogent. I recommend that the paper to be accepted.

Thank you for your encouraging evaluation of our paper and your time in reviewing our work.

In summary, the paper is well written, with clear method descriptions which would allow for other researchers to reproduce the approach. The approach will be helpful to biology researchers in a wide variety of domains searched with evolution algorithm? The core algorithms of mapping are open sourced. How about the registration algorithm? If it is open sourced, then a citation to where it can be found would be very helpful.

Thank you for this. We believe in making all our work open sourced whenever possible. The registration algorithm is now published: “Intensity-based registration of bright-field and second-harmonic generation images of histopathology tissue sections. Adib Keikhosravi, Bin Li, Yuming Liu, and Kevin W. Eliceiri. Biomedical Optics Express Vol. 11, Issue 1, pp. 160-173 (2020) •<https://doi.org/10.1364/BOE.11.000160>”. As mentioned in that publication, the registration algorithm is incorporated in our open-source collagen quantification tool “CurveAlign” that is freely available at <https://github.com/uw-loci/curvelets> and a validation of this algorithm is provided as well at https://github.com/uw-loci/shg_he_registration. In the revised manuscript, we have added the citation to this publication and updated the text accordingly.

Reference is well researched. There are some more recent work that would be helpful for the reader so the authors may consider adding them:

Bera, K., Schalper, K.A., Rimm, D.L. et al. Artificial intelligence in digital pathology — new tools for diagnosis and precision oncology. Nat Rev Clin Oncol 16, 703–715 (2019) doi:10.1038/s41571-019-0252-y

Xu, Z., Moro, C. F., Bozóky, B. & Zhang, Q. GAN-based virtual re-staining: a promising solution for whole slide image analysis. ArXiv.org <https://arxiv.org/abs/1901.04059> (2019).

Ihle, S.J., Reichmuth, A.M., Girardin, S. et al. Unsupervised data to content transformation with histogram-matching cycle-consistent generative adversarial networks. Nat Mach Intell 1, 461–470 (2019) doi:10.1038/s42256-019-0096-2

Thank you for this, we have now added these references to the introduction of the paper.

Reviewer #2 (Remarks to the Author):

In overall, the manuscript proceeds logically and reads quite well. I'm also quite convinced of the presented results and think that this work presents a significant advance in the field.

Thank you for your time and positive evaluation of our paper.

In order to further improve this high-quality work, I suggest the authors to consider the following points:

1. The sections regarding the algorithm and analysis details ('Registration based on mutual information maximization', 'Residual blocks' and 'Pixel shuffle layer') do not read as well as the other sections. I suggest to clarify and polish these sections further so that a reader would get a clearer picture of the presented approach, and would be able to better reproduce the results. It would also be useful to provide some references regarding the math related to Eq. (1).

Thanks for your comment, we have improved and added additional text to further clarify the sections regarding 'Residual blocks' and 'Pixel shuffle layer'. The details regarding the registration part has been recently published in a separate publication in Biomedical Optics Express journal and the citation to this new publication has been added in the revised manuscript.

2. The caption of Fig. 1. is not very clear/helpful to a reader. I suggest to clarify the overall workflow using a sentence or two.

Thank you for this comment, we have now clarified the figure caption as suggested.

3. Synthesis of second-harmonic generation (SHG) images is perhaps overly emphasized, because of which I fear that less careful readers may not fully appreciate the main novelty of this work. In my point of view, the important/interesting information that is extracted from the H&E images is related to the collagen organization. I understand that this information is often accessed by performing SHG microscopy, motivating to synthesize SHG images from the H&E images. However, from the biomedical point of view the SHG images as such are not important, rather the information they provide (in an easily accessible form). My suggestion is that the authors re-consider how much they want to emphasize the importance of SHG modality here. I think the impact of this work could be further increased by clarifying this point in the manuscript.

Thanks for your comment, we completely agree with your very good point. SHG is a very familiar and everyday term in our field that often times equates for collagen imaging and we forgot it may not be the same for many readers. Based on your suggestion we have now modified the text thoroughly to put the emphasis on fibrillar collagen itself rather than SHG, which is the current gold standard modality for label free imaging of this protein.

4. Line 169, Correct 'Kohler' to 'Koehler' or 'Köhler'.

Thank you for this correction, it has now been updated to Koehler.

5. Lines 187-188: I think a singular use of the reference [68] is adequate in this sentence.

Thank you for the comment, it has now been updated.

6. Line 448: 'Collagen crystals', you mean highly-ordered collagen fibers etc.?

Thank you for the comment, this has been updated.

7. Line 473: Clarify to the readers what the values 0.86 and 0.78 are and signify in this sentence. I don't think their meaning is obvious.

Thank you for the comment. In the revised manuscript, the definition and interpretation of the values, which are the Pearson correlation coefficient, have now been added in the manuscript.

8. Lines 490-495: Unnecessarily long sentence.

Thank you for the comment, it has been rewritten in the manuscript.

Reviewer #3 (Remarks to the Author):

From an engineering perspective the work and the presented images are impressive. The main limitation is that while CNN avoids the need for SHG imaging to evaluate collagen (which is extremely expensive) or specialized stains such as picosirius red, it still depends on having a high resolution whole slide image, or at least high resolution images of the entire diagnostically relevant regions of the tumor stroma. This is problematic, since at my center, images of breast and other surgical pathology specimens are never scanned, and in fact no whole slide imaging scanner is even present in the facility. If individual fields of tumor pathology could be selected, that would simplify data acquisition, but it would still force purchasing microscope hardware with the correct specifications (NA, sensor color filters, etc) to match the training set used on the CNN as well as training of personnel to select the correction locations to feed the CNN. This is at odds with the argument in the manuscript that the proposed CNN method could generate collagen alignment data without specialized equipment, at least as long as slide scanning remains uncommon. More likely, whole tissue slides would have to be sent to a third party service to be imaged on the correct hardware and then evaluated with the CNN. While slides are often sent to third parties to perform complicated or specialized diagnostics (e.g. FISH, etc) during routine workup of tumor specimens, the paper would benefit from a stronger argument that the CNN is the most practical approach compared to just having the third party do IHC or other special staining (or even SHG if only a small region of the tumor needed to be sampled).

Thanks for your comment and raising an important issue. Here we try to elaborate on the matter:

To the best of our knowledge and based on publications cited in the manuscript, stromal collagen reorganization is part of and a facilitator for tumor metastasis adjacent to the tumor boundary. Although there is possibility of relevant collagen biomarkers in distal areas, but we have no proof as of now, which in part could be due to the fact that SHG imaging and analysis on large areas and on a population of samples is very time and resource consuming. To apply current findings regarding stromal biomarkers in diagnostic pathology (such as example in figure 4), a single field of view of a few hundred microns would be sufficient that can be easily collected with a high magnification in one or few single shots. So, a regular camera (would not cost more than few hundred dollars) or even a cellphone camera can easily capture the image to feed into the CNN and achieve satisfactory results. So no more histochemical staining or SHG imaging is needed.

It's also worth mentioning that our group will be releasing a new version of the open source micromanager acquisition system with the support for color cameras, large stitching of image grids and autofocusing that is totally free, open source and capable of converting any microscope into a slide scanner just by adding a cheap color camera. We believe this would be a major step for collecting whole slide scans and looking for distal collagen biomarkers and pathologists would be easily able to incorporate into their everyday practice. We are also developing and have already published several publications on single-image super resolution algorithms for histopathological images that could enable researchers to even collect low magnification images and reconstruct higher resolution features to further lower the cost and time of imaging.

Finally, one thing that was less clear to me is how the authors envision collagen imaging (or any kind) being integrated into clinical practice. Would this be something that a pathologist would be trained to interpret, similarly to how special stains are used today? Or would it be a purely digital method where a CNN generates the images, and then an additional algorithm grades the specimen? The latter would strengthen the argument for the CNN approach since it would anyway require whole slide imaging.

This is a great comment and has been a concern of us as well and we are taking steps towards solving the issue. As we reviewed in the manuscript, the main collagen related biomarkers that have been explored and proven diagnostic so far are alignment of fibers and their orientation with respect to the tumor boundary. However, as you know based on your clinical experience, pathologists are less likely to incorporate these highly informative biomarkers in their everyday practice as they are biased towards pathological features related to cells and for example their differentiation. Also, we think there might be some patterns other than these simple biomarkers that have not been explored. To analyze the entirety of the tissue (both cellular and stromal features) we have trained deep CNNs that can be used for diagnosis and prognosis of pancreatic adenocarcinoma with higher accuracy than traditional methods that only use cellular features. A paper on this work is now being revised for submission. Further investigation of this method by us and other researchers would help the pathologist to find high risk patients by wholistic analysis of the tissue without even the need to extract the collagen image and analyze for finding diagnostic and prognostic biomarkers. However, this will need to be investigated for different disease independently on large datasets.

Minor:

Figure 4: Are the CurveAlign data here the orientations described in the methods section? If so, a colorbar with the scale would be helpful.

Thank you for this good suggestion, we have now added this.

“CNBs from prophylactic mastectomies of non-diseased women”

What was being biopsied if the patient was receiving a prophylactic mastectomy? Or were the biopsies performed on a mastectomy specimen (after surgery)?

Thank you for this very good question. The biopsies were done while the breast was still in place but just before the surgery started – the patient was anesthetized. We have now added this to the manuscript.

“A MIRA 900 Ti: Sapphire laser (Coherent, Santa Clara, CA) tuned to 780 nm, with a pulse length of approximately 100 fs”. The femtosecond model is the 900F, and I don’t think it can produce a pulse as short as 100fs. The actual length is probably somewhat longer.

Thank you for the correction, the text has been updated and highlighted to “less than 200 fs”

“an interference filter centered at 390 nm with a full width at half maximum bandwidth of 22.4 nm (Semrock)” Should give the model number.

Thank you for the correction, we appreciate this comment as it was a mistake and the filter is a Thorlabs MF390-18 and has now been corrected in the manuscript as well.

“imaged onto the 5 mm aperture of a 7422-40P photomultiplier tube” The model is actually the “H7422-40P”.

Thank you for the correction, it has been updated to H7422-40P

“(An example of selected ROIs is illustrated in Figure 6A and 6B).”

Figure 6 panels should probably be labeled A, B, ... to remove ambiguity.

Thank you for the comment, figure and text have now been updated and highlighted accordingly.

“Figure 5C and 5D.”

I think this actually meant to refer to Figure 6.

Thank you for the correction, the text has now been updated and highlighted accordingly.

REVIEWERS' COMMENTS:

Reviewer #1 (Remarks to the Author):

Thanks for taking all the suggestions from the reviewers and improving the manuscript accordingly. It is especially valuable to the community that the tools and the algorithms are open sourced and available for other researchers to validate and extend. I have no further suggestions and I look forward to sharing this article with colleagues when it's published.

Reviewer #2 (Remarks to the Author):

I think the authors have done a good job of addressing the concerns and questions raised by the reviewers. I think this manuscript is now a very solid piece of work that should be published.

Reviewer #3 (Remarks to the Author):

I am satisfied with the improvements to the manuscript and continue to think this is an impressive work.